# A Simple and Effective Method to Concentrate Hepatitis C Virus: Aqueous Two-Phase System Allows Highly Efficient Enrichment of Enveloped Viruses

**DOI:** 10.3390/v14091987

**Published:** 2022-09-08

**Authors:** Heesun Kim, Johan Yi, Jinbae Yu, Jaesung Park, Sung Key Jang

**Affiliations:** 1Molecular Virology Laboratory, POSTECH Biotech Center, Pohang University of Science and Technology, Pohang 37673, Korea; 2Nanoparticle and Vesicle Laboratory, Department of Mechanical Engineering, Pohang University of Science and Technology, Pohang 37673, Korea; 3Molecular Virology Laboratory, Department of Life Sciences, Pohang University of Science and Technology, Pohang 37673, Korea; 4Nanoparticle and Vesicle Laboratory, School of Interdisciplinary Bioscience and Bioengineering, Pohang University of Science and Technology, Pohang 37673, Korea

**Keywords:** hepatitis C virus, concentration, ultracentrifugation, aqueous two-phase system

## Abstract

To investigate the proliferation cycle of a virus, virus-host interaction, and pathogenesis of a virus, virion particles must be concentrated from the media of virus cell culture or the sera of virus-infected patients. Ultracentrifugation of the culture media is a standard method for concentrating virion particles. However, this method is time-consuming and requires special equipment (ultracentrifuge). Moreover, a large number of infectious viruses are lost during enrichment. We developed a new method of hepatitis C virus (HCV) concentration to overcome the issues associated with traditional methods of virus concentration. We used an aqueous two-phase system (ATPS) to concentrate the virus. HCV, which causes various liver diseases, such as liver fibrosis, cirrhosis, and hepatocellular carcinoma, was used as a model virus to test the efficacy and reliability of the ATPS. The efficiency of HCV concentration by the ATPS was approximately three times higher than that by ultracentrifugation. Moreover, the infectivity of the concentrated HCV, which is a labile virus, remained the same after concentration of the virus by the ATPS. Considering the simplicity and effectiveness of the ATPS, it is the method of choice for concentrating viruses.

## 1. Introduction

The concentration and/or purification of viruses is essential to investigate the basic aspects related to the viral life cycle, viral morphology, virus-host interaction, pathogenesis, immunity, and functions of viral proteins, as well as application aspects in the production of vaccines and the development of anti-viral drugs [1]. For instance, a high-titer virus (multiplicity of infection ≥ 5) is required to infect more than 99% of cells in order to study one-step growth curves of a specific virus [2]. Currently, the canonical methods for concentrating viruses are rate-zonal centrifugation in the presence of sucrose [3], isopycnic centrifugation in the presence of cesium chloride [4,5], polyethylene glycol (PEG) precipitation [3], and centri-filtration [6,7]. However, these virus concentration methods have limitations, such as low yield, virus destabilization, long experimental time, and requirement of expensive equipment [8].

Hepatitis C virus (HCV) is an enveloped virus that contains a positive-sense single-stranded RNA genome. HCV belongs to the *Hepacivirus* genus in the *Flaviviridae* family [9]. Globally, 71 million patients suffer from chronic HCV infection, which causes various liver diseases, such as liver fibrosis, chronic hepatitis, liver cirrhosis, and hepatocellular carcinoma [9]. Very effective direct-acting antiviral drugs are available for the treatment of HCV [10], but a vaccine for preventing HCV infection has not yet been developed [11].

In general, the density of HCV particles circulating in patient sera varies because virion particles are associated with lipid molecules and lipoproteins, such as ApoE and ApoB, which are components of very-low-density lipoproteins [12]. The interaction between HCV and lipoproteins forms a lipo-viro-particle containing viral RNA, a capsid protein core, and viral envelope proteins E1 and E2. The association of lipids and lipoproteins with HCV results in varying sizes of virion particles, ranging from 40 to 100 nm in diameter [13].

The general methods for the concentration of HCV particles are precipitation by ultracentrifugation, precipitation with PEG, and filtration [3,4,5,6,7]. The most frequently used method is the precipitation of HCV particles using an ultracentrifuge in the presence of a sucrose gradient or sucrose cushion [14]. Although ultracentrifugation is the most commonly used method to concentrate viruses [15,16], the enrichment of viruses by ultracentrifugation has several drawbacks. In particular, the recovery yield is low. For instance, the recovery rates of MS2 virus and hepatitis A virus after concentration by ultracentrifugation are only 5% and 9%, respectively [17,18]. Moreover, the precipitation of virion particles with an ultracentrifuge is time consuming, and the high centrifugal force (~100,000× *g*) applied to the virion particles may damage them. PEG precipitation is used to enrich viruses from a large volume of virus stock [3]. However, a large portion of virion particles remains in the aqueous solution after precipitation and can be damaged in the process of dissolving the precipitated virion particles. Centri-filtration with a cut-off filter is possible, but is not cost-effective [6].

The aqueous two-phase system (ATPS) consisting of a polymer of dextran/PEG is widely used for bioparticle concentration [19]. This system uses a much lower gravitational force (~1000× *g*) than ultracentrifugation (~100,000× *g*) to isolate the bioparticles. It has been shown that extracellular vesicles (50–200 nm), with similar sizes to many viruses, can be efficiently enriched by an ATPS [20,21]. However, the possibility of using an ATPS (dextran/PEG) to concentrate an enveloped virus has not yet been tested. Here, we report a new method for concentrating enveloped viruses using an ATPS. As a model virus, we used HCV, which is an enveloped labile virus. The ATPS was found to be a much more effective method than the conventional ultracentrifugation method for concentrating virion particles. Moreover, HCV infectivity was not affected by ATPS concentration. Considering the effectiveness, running time, and cost, the ATPS is a much better method than conventional methods for concentrating enveloped viruses.

## 2. Materials and Methods

### 2.1. Cell Lines and Cell Culture

Huh-7.5.1 cells were cultured in Dulbecco’s Modified Eagle’s Medium (Gibco^TM^) containing antibiotics (100 U/mL penicillin and 10 μg/mL streptomycin) supplemented with 10% fetal bovine serum (FBS, PEAK, Inc., Wellington, CO, USA) in a humidified CO_2_ (6.0%) incubator at 37 °C.

### 2.2. Virus Production

In-vitro transcribed HCV RNAs were electrophoresed in Huh-7.5.1 cells as described previously [22]. Two RNAs that produce two different variants of HCV were used in this study. The first was JC1-E2-Flag, which is a derivative of JC1 with the insertion of a flag tag between the E1 and E2 sites [22]. The JC1-E2-Flag variant was used for measuring the concentrations of nanoparticles and viral RNAs, visualizing the morphology of HCV, and determining the focus forming units (FFUs) of HCV. The other was JFH-5a Rluc, which is a derivative of JFH1 with cell culture adaptive mutations in the E2 and p7 regions and the *Renilla* luciferase gene for convenient monitoring of HCV proliferation [23]. The RNA-transfected cells were cultivated for 7 days in extracellular vesicle-depleted FBS (d-FBS) medium prepared using ultracentrifugation at 100,000× *g* at 4 °C for 20 h. Culture media containing virion particles (HCVcc) were collected from 3 to 7 days after transfection and filtered through a polyether sulfone (PES) filter (0.45 μm). HCV particles were concentrated from the media using either ultracentrifugation or an ATPS.

### 2.3. Ultracentrifugation

HCV culture media were loaded onto 20% sucrose cushions in TNE buffer (100 mM NaCl, 5 mM Tris-HCl (pH 8.0), 1 mM EDTA), and HCV particles were precipitated by ultracentrifugation at 100,000× *g* at 4 °C for 2 h (45 Ti rotor; Beckman, Brea, CA, USA). The pellets were resuspended in d-FBS medium and filtered through syringe filters (0.20 μm) for further analyses.

### 2.4. Aqueous Two-Phase System (ATPS)

The ATPS was prepared from a mixed solution of dextran (Molecular weight (MW) 35,000–45,000, Sigma Aldrich, St. Louis, MO, USA) and PEG (MW 25,000–45,000, Sigma Aldrich). Fifty milliliters of 3X ATPS solution (3.9% *w*/*w* of dextran and 13.5% *w*/*w* of PEG in phosphate buffered saline) was mixed with 100 mL of HCV culture medium. The mixture of the ATPS solution and HCV culture medium (150 mL) was dispensed into four conical tubes (37.5 mL each). The mixture was centrifuged at 1000× *g* for 10 min to form two phases. The upper phase containing PEG was discarded, and the dextran phase containing concentrated HCV was collected. The dextran phase was diluted with d-FBS medium to a final volume of 2 mL and then filtered through a syringe filter (0.20 μm) for further analyses.

### 2.5. Characterization of Viruses

HCV particles were characterized using transmission electron microscopy (TEM) and nanoparticle tracking analysis (NTA) (Exocope, Exosomeplus). For TEM, samples were loaded onto formvar-coated grids (Electron Microscopy Sciences) and negatively stained with UranyLess (Electron Microscopy Sciences, Hatfield, PA, USA) for 1 min at room temperature. The samples on the grids were observed using TEM (Jeol). For NTA, HCV solutions were diluted (100 pts/field). NTAs were performed at different concentrations (at least six times each) for quantitative comparison.

### 2.6. Quantification of HCV RNA

HCV RNAs were purified using Trizol LS Reagent (Ambion, Austin, TX, USA), according to the manufacturer’s protocol. Complementary DNA synthesis was performed using a reverse transcription reagent (Promega, Madison, WI, USA). Quantitative reverse transcription polymerase chain reaction (qRT-PCR) was performed using the TaKaRa SYBR Premix EX Taq II protocol with an IQ5 multicolor real-time PCR detection system (Bio-Rad, Hercules, CA, USA). The sequences of primers for qRT-PCR were as follows: 5′-TCTGCGGAACCGGTGAGTA-3′ and 5′-TCAGGCAGTACCACAAGGC-3′.

### 2.7. Immunocytochemistry

Huh-7.5.1 cells were cultivated on 12-well plates coated with poly-L-lysine (Sigma Aldrich, St. Louis, MO, USA) for 24 h, then inoculated with viruses for 4 h. Two days after infection, the cells were fixed for immunocytochemistry. The fixed cells were treated with 1% bovine serum albumin solution and then with an anti-HCV core antibody (mouse) (MA1-080, Thermo Fisher Scientific, Waltham, MA, USA) and 2′-[4-ethoxyphenyl]-5-[4-methyl-1-piperazinyl]-2,5′-bi-1H-benzimidazole trihydrochloride trihydrate (Hoechst 33342, Thermo Fisher Scientific, Waltham, MA, USA). An Alexa 555-labeled anti-mouse antibody (Thermo Fisher Scientific, Waltham, MA, USA) was used to visualize the HCV core protein. Fluorescence microscopy was performed on a Zeiss Axio Scope A1 microscope.

### 2.8. Measurement of Virus Infectivity

Huh-7.5.1 cells were inoculated with virus samples (JFH-5a Rluc). After 4 h of viral inoculation, the culture medium was replaced with fresh medium. The cells were further cultivated for 3 days. Cells were harvested and lysed in passive lysis buffer (Promega, Madison, WI, USA), and luciferase activity in cell lysates was measured using the *Renilla* Luciferase Assay System (Promega, Madison, WI, USA).

### 2.9. Efficiency of HCV Recovery

The recovery efficiencies of virion particles and viral RNAs after the concentration process by either ultracentrifugation or ATPS were calculated as follows:$$\mathrm{Particle}\;\mathrm{recovery}\;\mathrm{efficiency}\;\left(\%\right)=\frac{\mathrm{Particle}\;\#\;\mathrm{after}\;\mathrm{enrichment}\;\times \;\mathrm{final}\;\mathrm{sample}\;\mathrm{volume}}{\mathrm{Particle}\;\#\;\mathrm{before}\;\mathrm{enrichment}\;\times \;\mathrm{initial}\;\mathrm{volume}\;\mathrm{of}\;\mathrm{HCV}\;\mathrm{media}}\times 100.$$
$$\mathrm{RNA}\;\mathrm{recovery}\;\mathrm{efficiency}\;\left(\%\right)\;\;\;\;\;\;\;\;\;\;\;\;\;\;\;\;\;\;\;\;\;\;\;\;\;\;\;\;\;\;\;=\frac{\mathrm{HCV}\;\mathrm{RNA}\;\mathrm{concentration}\;\mathrm{after}\;\mathrm{enrichment}\;\times \;\mathrm{final}\;\mathrm{sample}\;\mathrm{volume}}{\mathrm{HCV}\;\mathrm{RNA}\;\mathrm{concentration}\;\mathrm{before}\;\mathrm{enrichment}\;\times \;\mathrm{initial}\;\mathrm{volume}\;\mathrm{of}\;\mathrm{HCV}\;\mathrm{media}}\times 100.$$
$$\mathrm{Focus}\;\mathrm{forming}\;\mathrm{unit}\;\left(\mathrm{FFU}\right)\;\mathrm{recovery}\;\mathrm{efficiency}\;\left(\%\right)\;\;\;\;\;\;\;\;\;\;\;\;\;\;\;\;\;\;\;\;\;=\frac{\mathrm{HCV}\;\mathrm{FFU}\;\mathrm{after}\;\mathrm{enrichment}\times \mathrm{final}\;\mathrm{sample}\;\mathrm{volume}}{\mathrm{HCV}\;\mathrm{FFU}\;\mathrm{before}\;\mathrm{enrichment}\times \mathrm{initial}\;\mathrm{volume}\;\mathrm{of}\;\mathrm{HCV}\;\mathrm{media}}\times 100$$

## 3. Results

### 3.1. ATPS Is a Highly Effective Virus Concentration System (Comparison between ATPS and Ultracentrifugation)

We performed concentration of HCV using either an ATPS or a conventional ultracentrifugation method (Figure 1). To determine the virus enrichment efficiencies of the ATPS and ultracentrifugation, we performed NTA (Figure 2), quantification of viral RNA (Figure 3), and measurement of the infectious dose of enriched viruses (Figure 4). Through the NTA method, the number and size of individual nanoparticles can be measured from the Brownian motion of the individual particles (Figure 2c). To compare the efficacy of enrichment, the solution volumes of the re-dissolved pellet after ultracentrifugation (Ultra-Pellet) and the ATPS dextran layer (ATPS-DEX) were adjusted to 2 mL each. Surprisingly, the enrichment of nanoparticles by the ATPS, including virion particles and extracellular vesicles (EVs), was approximately three-fold higher than that by the conventional ultracentrifugation method (Figure 2a). Moreover, about three-quarters of the nanoparticles in the culture media were enriched in the dextran layer of the ATPS (Figure 2b). In contrast, only a quarter of the nanoparticles in the culture media were enriched in the pellet after ultracentrifugation (Figure 2b). In other words, the majority of nanoparticles still remained in the supernatant after ultracentrifugation (Figure 2b).

### 3.2. The Size Distribution of Nanoparticles Enriched by ATPS and Ultracentrifugation

The sizes and amounts of nanoparticles enriched by the ATPS and ultracentrifugation were analyzed using NTA and TEM. The amounts of nanoparticles in the solutions before and after enrichment varied, but the sizes of the nanoparticles in the peaks were similar (approximately 100 nm) among the solutions (Figure 2c). The sizes of the enriched nanoparticles ranged from 50 nm to 200 nm. The broad range of enriched nanoparticles is likely due to the variable sizes of HCV particles that are associated with lipoproteins [12,13] and the variable sizes of EVs [24].

The nanoparticles in the unenriched and enriched samples were visualized using TEM (Figure 2d). Seeming EVs (indicated by white arrows) and HCV particles (indicated by solid arrows) were observed before and after the enrichment steps. Many more EVs and HCV particles, shown as spherical shapes [13,25,26], were observed after the enrichment processes (Figure 2d, first panel vs. second and third panels). The morphology of HCV particles did not change after the enrichment step (Figure 2d). We observed materials of new shapes (indicated by yellow arrows) that were detected in the dextran layer after the ATPS. We speculate that these particulate materials are very-low-density lipoproteins [27,28].

### 3.3. Quantification of HCV RNAs before and after Enrichment Processes

According to the NTA, three times more nanoparticles were enriched by the ATPS than by ultracentrifugation (Figure 2a, Appendix A). However, it was not clear how many HCV particles were enriched by ultracentrifugation and the ATPS, since both methods can enrich not only virion particles, but also EVs. We measured the amounts of HCV particles in each solution prepared by different methods using qRT-PCR. Two primers targeting HCV RNA were used in the PCR and reverse transcription reactions. RNA generated by in vitro transcription of JC1-E2-Flag was used as a control RNA for the quantification of purified viral RNAs from the enrichment steps.

The relative amounts of HCV RNAs in each fraction, quantified by qRT-PCR, are shown in Figure 3a. The amount of HCV RNAs in the dextran layer of the ATPS was about three times higher than that enriched by ultracentrifugation (Figure 3a, Appendix A, Ultra-Pellet column vs. ATPS-DEX column). The enrichment ratio of HCV RNAs by ATPS compared to ultracentrifugation (Figure 3a) was the same (approximately three-fold) as the enrichment ratio of nanoparticles by these two methods (Figure 2a). These results indicate that the enrichment of EVs and HCV particles occurs at similar levels using both the ATPS and ultracentrifugation. The efficacy of HCV RNA enrichment by the two methods was calculated as follows: (concentration of HCV RNA copies × final volume) ÷ (concentration of HCV RNA copies × initial volume of HCV-containing media). Approximately 40% of viral RNAs were enriched by the ATPS, but only approximately 15% of viral RNAs were enriched by ultracentrifugation (Figure 3b).

### 3.4. Determination of Viral Titers before and after Enrichment Processes

We monitored the infectivity of the concentrated HCV using HCV strain JFH5a-Rluc-ad34 containing a reporter gene (*Renilla* luciferase) in the NS5A region, which allows easy monitoring of viral infectivity by measuring *Renilla* luciferase activity in cells [23]. The concentrated viruses were inoculated into Huh-7.5.1 cells for 4 h. After changing the medium to a fresh medium, the cells were further incubated for 3 days. The cells were harvested and *Renilla* luciferase activity in the cell lysates was measured to monitor the infectivity of the virus. As shown in Figure 4a, the viral titer enriched by the ATPS was approximately 2.5 times higher than that by ultracentrifugation. In contrast, very low luciferase activity was detected in cells inoculated with the supernatant of ultracentrifugation and the PEG phase of the ATPS (Figure 4a). We also confirmed viral infectivity by western blotting using an antibody against NS5A of HCV (Figure 4b). Very strong NS5A bands were detected in samples from the ultracentrifugation pellet and the dextran phase of the ATPS, and a weak NS5A band was detected in the solution before enrichment (Figure 4b). Very weak NS5A bands were detected in samples from the supernatant of ultracentrifugation and the PEG phase of the ATPS (Figure 4b).

Finally, we confirmed the infectivity of samples before and after virus enrichment using an immunocytochemical technique [23] with a monoclonal antibody against the HCV core protein. HCV-infected cells are shown in red in Figure 4c. The focus forming units (FFUs) in samples were measured by counting the number of foci after fluorescence-labeling of HCV-infected cells using an Alexa 555-conjugated secondary antibody against mouse immunoglobulin G. The total FFUs in the samples were calculated and are shown in Figure 4d. Approximately 40% of infectious viruses were enriched by the ATPS method, but only 11% of infectious viruses were enriched by the ultracentrifugation method (Figure 4d). This indicates that the ATPS provides results 3.6 times better than ultracentrifugation in the enrichment of infectious HCV.

## 4. Discussion

Attempts have been made to use ATPSs to enrich viruses and virus-like particles (VLPs). For instance, adenovirus was enriched from the crude lysate of HEK293 cells using a PEG 300-phosphate system [29]. In addition, ATPSs have also been used for the purification of VLPs of rotavirus and human B19 parvo [30,31]. However, the utilization of ATPSs for purifying viruses and VLPs has not been fully explored because of the complex partitioning mechanism. Moreover, all the viruses and VLPs described above are non-enveloped viruses or VLPs of non-enveloped viruses. Enrichment of an enveloped virus by ATPSs remains unexplored. The outermost layer of non-enveloped viruses is composed of capsid proteins, whereas that of enveloped viruses consists of cell membranes originated from host cells. Therefore, because the biophysical properties of the outermost layers determine the behavior of nanoparticles in an ATPS, the enveloped and non-enveloped viruses most likely behave completely differently in various ATPSs. Recently, an ATPS was reported to concentrate EVs from cell culture media [32]. EVs were shown to be enriched in the dextran phase via hydrophobic and hydrophilic interactions [32]. Here, we investigated whether the conditions of the ATPS used in enriching EVs are applicable for the enrichment of an enveloped virus, HCV, which is a typical enveloped virus of varying sizes.

The recovery rate of HCV RNA by ultracentrifugation was approximately 15% (Figure 3b). The recovery rate of HCV infectivity after ultracentrifugation was further reduced by 25% compared to that of HCV RNA (15% vs. 11%) (Figure 3b and Figure 4d). Putative damage to virion particles by the high gravitational force (100,000× *g*) applied during sedimentation and the dissolving process of viral pellets is a likely reason for the further reduction of virus infectivity after ultracentrifugation. On the contrary, the recovery rate of HCV RNA by the ATPS (~40%) was 2.7 times higher than that by ultracentrifugation (Figure 3b). The recovery rate of HCV infectivity was approximately 40% (Figure 4d), which was the same as the recovery rate of HCV RNA (Figure 3b). These results indicate that the recovery rate of virus enrichment by the ATPS was much higher (2.7-fold and 3.6-fold in regard to viral RNA and virus infectivity, respectively) than that by ultracentrifugation, and that there was no loss of virus infectivity due to virus damage when the ATPS method was used.

As shown above, the ATPS was found to be a much more effective way to concentrate virion particles of an enveloped virus HCV than the conventional virus concentration method of ultracentrifugation. However, a few issues remain unresolved. First, approximately 2% of dextran existed in the dextran phase of the ATPS, where the virion particles were enriched. We tested the effects of dextran on HCV infectivity up to 4%, and no effect was found (Appendix A). These results indicate that the ATPS can be used for the enrichment of HCV for various purposes. Notably, some sucrose remained after the enrichment of virion particles by ultracentrifugation. Second, virion particles and EVs were enriched in the dextran phase of the ATPS. However, the enrichment of EVs is also unavoidable when using the ultracentrifugation method. Additional enrichment methods, such as filtration and affinity column purification, may be used for the further purification of viruses.

In conclusion, we developed a quick, easy, and highly effective method for enriching an enveloped virus HCV using an ATPS. Considering that the same ATPS works well for the concentration of both HCV (this study) and EVs [32] which contain very different protein and other internal components except the outer membranous structures (lipid bilayers), we speculate that this method could potentially be used for the enrichment of many enveloped viruses for academic purposes, as well as for biotechnologies related to diagnosis and vaccine production.

## Figures and Tables

**Figure 1 viruses-14-01987-f001:**
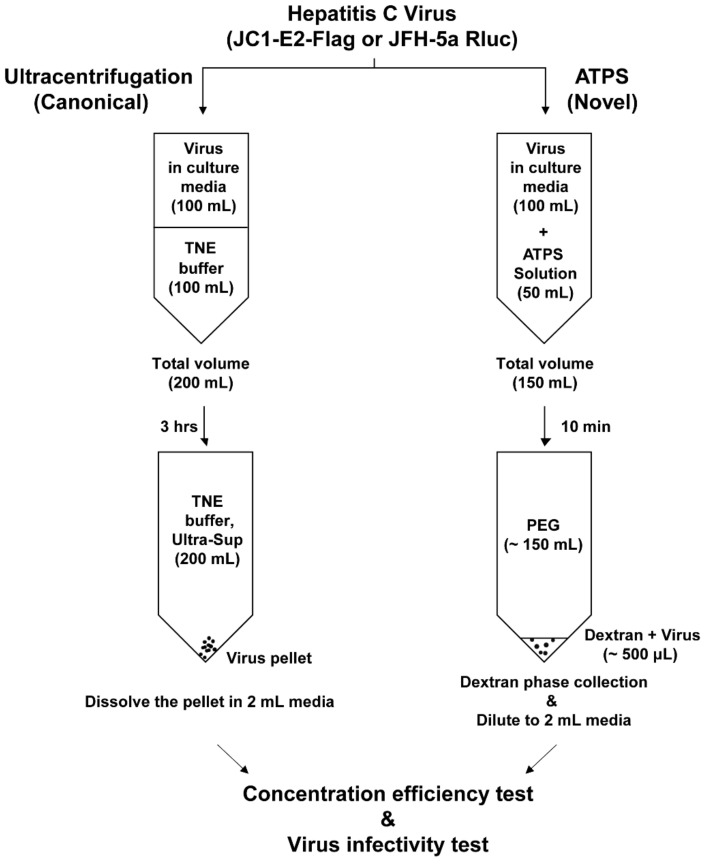
A schematic diagram of hepatitis C virus (HCV) enrichment by ultracentrifugation and the aqueous two-phase system (ATPS). HCV culture media (100 mL) were used in the enrichment processes using either ultracentrifugation (**left** panel) or the ATPS (**right** panel). **Left** panel: The HCV culture medium (100 mL) was loaded on top of TNE buffer containing 20% sucrose and then centrifuged at 100,000× *g* at 4 °C for 3 h. After centrifugation, the pellet was dissolved in 2 mL of culture medium. **Right** panel: The HCV culture medium (100 mL) was mixed with 3 × ATPS mixture (50 mL) containing dextran and polyethylene glycol (PEG) in phosphate-buffered saline. ATPS was performed by centrifuging at 1000× *g* at 4 °C for 10 min. After the enrichment process, the sample in the dextran phase was diluted with culture medium to a final 2 mL.

**Figure 2 viruses-14-01987-f002:**
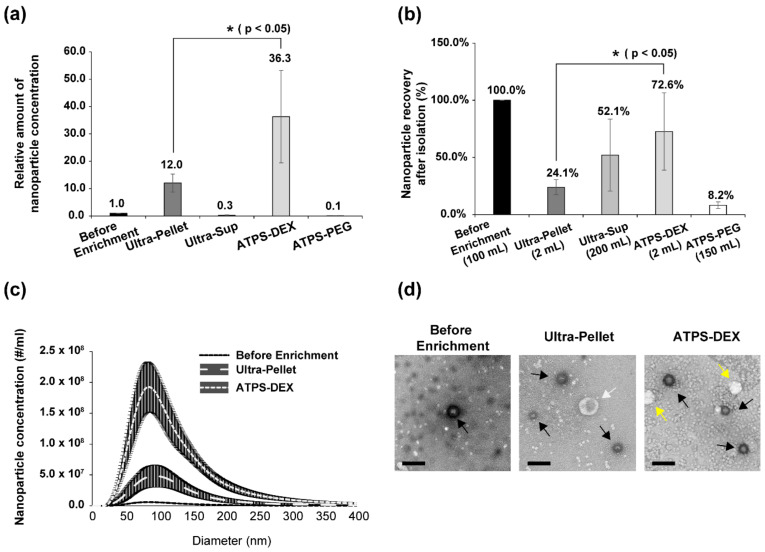
An analyses of nanoparticles enriched by ultracentrifugation and the aqueous two-phase system (ATPS). (**a**) The relative nanoparticle concentration was measured using nanoparticle tracking analysis (NTA). HCV (JC1-E2-Flag variant) particles were enriched by either ultracentrifugation or the ATPS. (**b**) Efficiencies of virus recovery were calculated using the concentration and volume of each fraction. (**c**) Size distribution of nano-sized particles was analyzed using NTA before and after enrichment. The concentration and size of nanoparticles were calculated by tracking the Brownian motion of particles. Experiments were performed three times with six times of particles tracking for each experiment. The shaded areas represent standard deviations. (**d**) Morphology of virus particles determined using transmission electron microscopy (TEM). HCV samples were negatively stained and observed using TEM. TEM revealed seeming HCV particles in spherical shapes. Black, white, and yellow arrows depict seeming HCV particles, extracellular vesicles, and lipoproteins (very-low-density lipoproteins), respectively (scale bar = 200 nm). The columns and bars in panels (**a**,**b**) represent the means and standard deviations, respectively. The average value is depicted on the top of each column. *p* values less than 0.05 are indicated by one asterisk (*). ATPS-DEX: ATPS dextran layer; ATPS-PEG: ATPS polyethylene glycol (PEG) layer.

**Figure 3 viruses-14-01987-f003:**
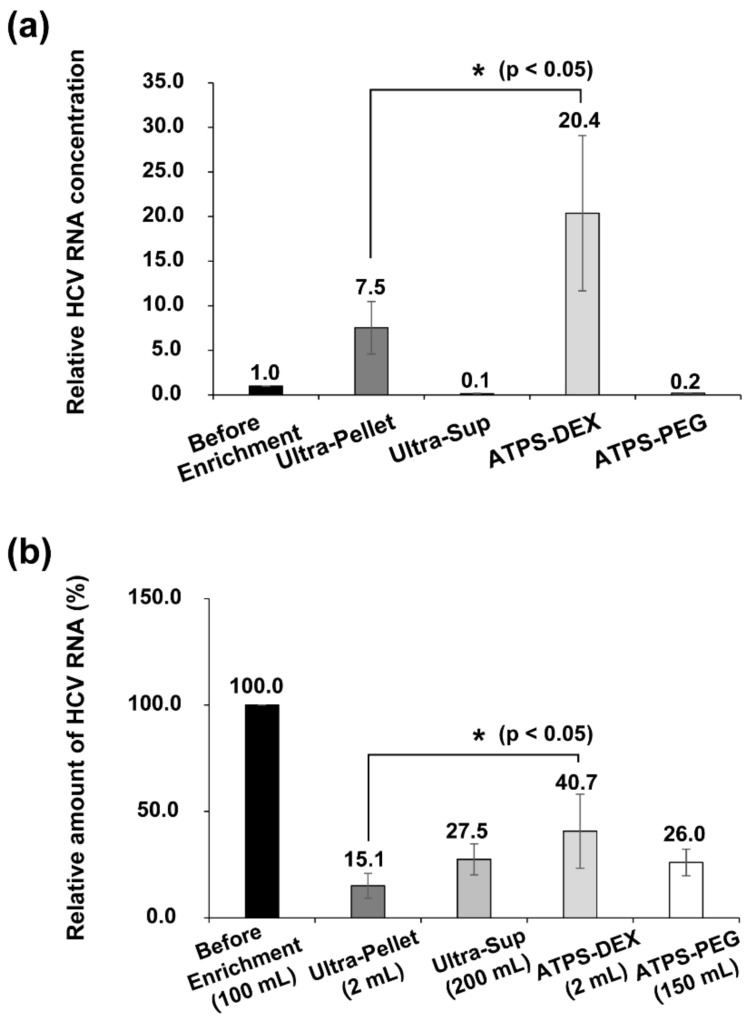
Efficacies of the hepatitis C virus (HCV) RNA enrichments by ultracentrifugation and the aqueous two-phase system (ATPS). The HCV (JC1-E2-Flag variant) RNAs in samples before and after enrichment processes (ultracentrifugation and ATPS) were quantified using quantitative reverse transcription polymerase chain reaction. (**a**) The relative concentrations of HCV RNAs in solutions before and after enrichment are depicted. (**b**) The recovery rates of the enrichment processes were calculated as described in the Materials and Methods. The columns and bars represent the means and standard deviations, respectively. The average value is depicted on the top of each column. *p* values less than 0.05 are indicated by one asterisk (*). Experiments were performed three times. ATPS-DEX: ATPS dextran layer; ATPS-PEG: ATPS polyethylene glycol (PEG) layer.

**Figure 4 viruses-14-01987-f004:**
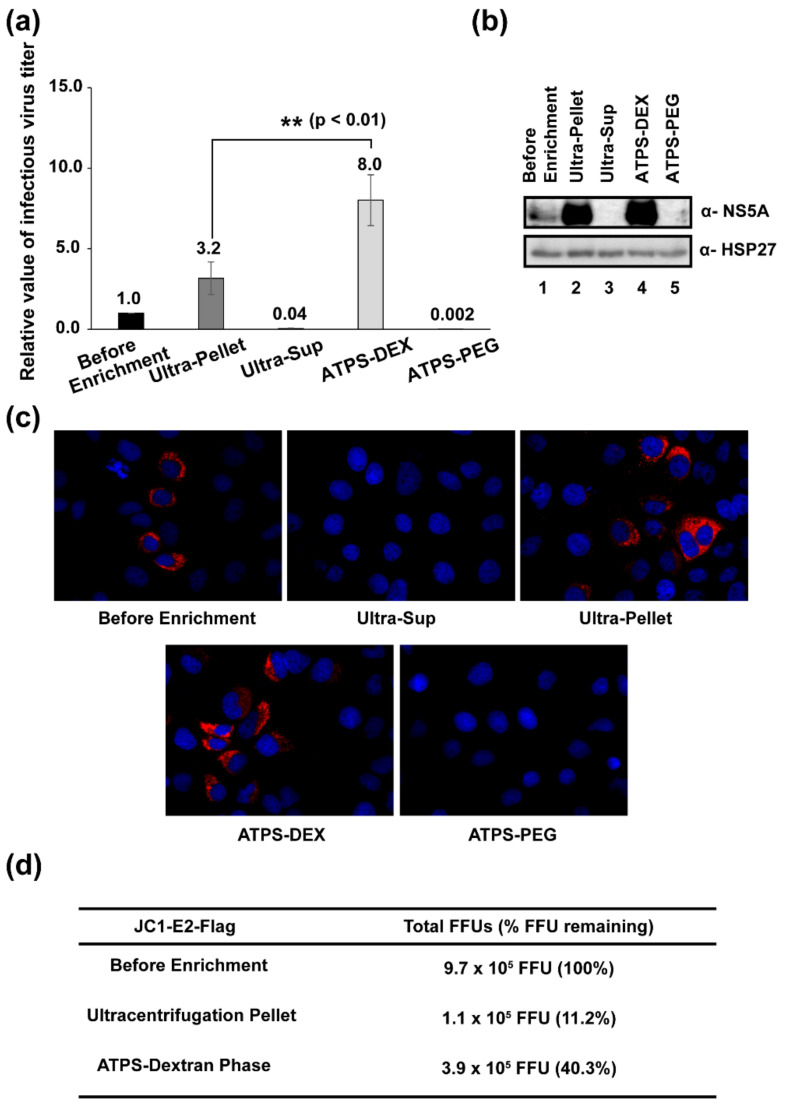
The infectivity of hepatitis C virus (HCV) particles enriched by ultracentrifugation and the aqueous two-phase system (ATPS). (**a**) The Huh-7.5.1 cells were inoculated with the virus (JFH-5a Rluc variant) solutions, and the proliferation levels of HCV were monitored by measuring luciferase activities in the cell extracts. Experiments were performed three times in duplicate. The columns and bars represent the means and standard deviations, respectively. The average value is depicted on the top of each column. *p* values less than 0.01 are indicated by two asterisks (**). (**b**) Monitoring of HCV (JFH-5a Rluc variant) proliferation by western blotting. The levels of the HCV NS5A in the infected cells were monitored by western blotting to observe HCV protein production. (**c**) Immunocytochemistry of HCV (JC1-E2-Flag variant)-infected cells. The HCV core proteins (shown in red) were observed in HCV-infected cells with JC1-E2-Flag. The nuclei of cells (shown in blue) were visualized using 2′-[4-ethoxyphenyl]-5-[4-methyl-1-piperazinyl]-2,5′-bi-1H-benzimidazole trihydrochloride trihydrate. (**d**) The focus forming units (FFUs) measured by immunocytochemistry (panel c) are depicted in the chart. The recovery rates were calculated as described in Materials and Methods. ATPS-DEX: ATPS dextran layer; ATPS-PEG: ATPS polyethylene glycol (PEG) layer.

## Data Availability

Not applicable.

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
