# Peer review of "A Simple and Effective Method to Concentrate Hepatitis C Virus: Aqueous Two-Phase System Allows Highly Efficient Enrichment of Enveloped Viruses"

_viruses, 2022, doi:10.3390/v14091987_

Round 1

Reviewer 1 Report (Previous Reviewer 1)

The authors have properly addressed my criticism(s) and the paper is now acceptable for publication.

Author Response

Reviewer 2 Report (New Reviewer)

The authors use a simple and cost effective protocol initially established for the concentration of extracellular vesicles to concentrate cell culture derived HCV from the supernatant of infected cells, as a proof of concept for the applicability towards enveloped viruses. In my view, this paper could be a great resource, inspiring many virologists to give it a try with their virus of interest. I was not engaged in the first reviewing round, but most of the reviewer comments criticized widely phrased claims based on limited datasets. Those issues have been addressed by rephrasing the claims and not by adding the requested data. Some more data asked for by reviewer 2 would have been nice, e.g. enrichement of HCV from serum, although in this case infectivity and integrity of particles would have been hard to demonstrate. Addition of another enveloped virus would have been nice as well, still the way how the authors handled the concerns is acceptable, although probably not preferable.

Since I entered the reviewing process only at this late stage, I am limiting additional concerns to the EM dataset, which in my view requires attention.

Specific point:

The authors claim to show HCV particles by TEM upon negative staining. Demonstrating HCV particles in EM is notoriously difficult, other studies have provided intensive efforts to validate the nature of the vesicles as HCV virions in previous studies, due to the lack of distinct substructures allowing a clear discrimination from extracellular vesicles. The strongly stained material could simply be damaged vesicles, allowing the stain to enter. Since both processes will enrich EVs to the same extent as virions, increased abundance cannot be taken as an argument. Since the authors use flag-tagged E2, they could simply validate the nature of the vesicles by immuno-EM, which I would recommend. Otherwise they need to tone down the claim that the marked structures really represent authentic virions.

Author Response

This manuscript is a resubmission of an earlier submission. The following is a list of the peer review reports and author responses from that submission.

Round 1

Reviewer 1 Report

The manuscript by Kim and coauthors reports that aqueous two-phase system, or ATPS, can be effectively and readily utilized to prepare concentrated stocks of an enveloped virus (HCV) while maintaining virtually intact its infective potential. The paper is interesting, technically sound and clearly written. The conclusions that the authors reach ("we developed a quick, easy, and highly effective method for enriching enveloped viruses using an ATPS. This method can be used for the enrichment of many enveloped viruses for academic purposes, as well as for biotechnologies related to diagnosis and vaccine production"), however,  is not fully justified by the data. They showed that the ATPS methodology works for HCV. That is one virus only and it may very well be that the unique composition of HCV (a virolipoparticle) is what makes the procedure so efficient for this virus.  I agree that this work could have wide applications for a variety of different viruses, but this has yet to be demonstrated.

All in all, I do recommend publication of this paper in Viruses but only after the last sentence (quoted above) is rewritten and the limitations of the study are more clearly stated.

Reviewer 2 Report

In this paper the authors describe the application of an existing method to purification of two engineered variants of cell culture produced Hepatitis C virus.

Overall the study includes measurements that make sense and are excuted appropriately, however, the authors draw conclusions that are not supported by the results that are presented. 

This method is not a new method, as described in the conclusion, the method has been used to concentrate other viruses. 

The study mentions several times that this method can be used to concentrate viruses, however the authors only used two engineered variants fo HCV. Additionally, the authors mention in the conclusion that many viruses can not be concentrated using this method.

Please alter the conclusions to be more precisely in line with the limited findings in this study. 

- It is unclear which variants are used for which experiments. Are the authors only using the FLAG variant for most experiments, and the Luc variant for infectivity? Please clarify in the methods, and in the legend of each figure. 

- Why did the authors not choose to include the normal variant of HCV (JFH-1, or JC-1), instead of, or in addition to engineered viruses? 

- HCV in the patient population circulates in a large variety of densities, and these densities, due to association with lipoproteins, are largely underrepresented or absent in tissue culture produced HCV.

In order to strengthen the conclusion about HCV alone, I would suggest including both 'natural' HCV derived from patients, and un-engineered tissue culture-produced HCV. 

With the addition of these experiments the authors could potentially draw the conclusion that this method can be used to purify tissue culture produced HCV, and/ or natural occurring HCV.

if the authors want to draw conclusions about viruses in general, these experiments have to be repeated for a broad range of viruses, to better indicate what the limits are for the purification of viruses using this method.

- There are several typos in the manuscript. 

Reviewer 3 Report

This is a technical report in which the authors have compared the standard method to purify HCV particles by ultracentrifugation to an aqueous two-phase system. It remains essential to improve methods that not only allow concentration of viral particles but also preserve their most native conformation and their infectious potential, or allow their purification and removal of e.g., serum components. 

While this report focuses on an interesting and important topic, several points of the study needs to be addressed before this manuscript can be published by Viruses.

1 – Figure 2. It is important to provide quantitative results, not relative values of enrichment in panels a and b. Panel c: please provide repeat of the NTA assay. In panel d, the quality of the EM picture is very low: this cannot be published as such. Please provide images at better resolution and immuno-gold labeling of HCV antigens.

2 – Likewise in figures 3 and 4: please provide real numbers. Panel b of Figure 3: provide repeats and quantification analysis. 

3 – it is essential to provide data of the depletion factors of some serum/media components, while comparing ultracentrifugation vs. the aqueous two-phase system. Please titrate serum albumin, BSA and HSA, as well as other serum and cellular (Huh-7.5 cells) markers such e.g.  as transferrin, alpha-1-antitrypsin, apoE, apoB.